# Zero-Shot Cross-Domain Dialogue State Tracking with small LLMs: Learning to Think through Reinforcement Learning

## Abstract

Dialogue State Tracking (DST) is essential for task-oriented dialogue systems to track user goals, but zero-shot adaptation to unseen domains poses significant challenges. This paper proposes an innovative approach to enhance small LLMs for zero-shot cross-domain DST using reinforcement learning (RL) with verifiable rewards. We introduce two novel techniques: a Dynamic Difficulty Sampling Pipeline, which adaptively selects training examples to optimize learning efficiency, and a Difficulty-Weighted Fuzzy Match Reward Function, which provides granular feedback to address sparse rewards and prioritize difficult slots. Employing the Group Relative Policy Optimization (GRPO) algorithm, our method boosts the reasoning capabilities of small LLMs, enabling robust generalization to new domains without further training. Experiments on MultiWOZ 2.1 and 2.4 show our approach achieves state-of-the-art performance among small models and rivals larger ones, while being computationally efficient. This work demonstrates the effect of RL-based post-training for compact LLMs, paving the way for scalable, resource-efficient dialogue systems. Our code and model is available at (https://anonymous.4open.science/r/DSTRL-769B).

## 1 Introduction

Dialogue State Tracking (DST) is the process of maintaining a structured representation of user goals, often as slot-value pairs, to support effective dialogue management Budzianowski et al. (2018). This paper investigates the application of reinforcement learning (RL) with verifiable rewards Shao et al. (2024) to small large language models (LLMs) specifically for zero-shot cross-domain DST Williams et al. (2016), enabling the handling of unseen domains without additional training. By leveraging RL, we aim to enhance the reasoning capabilities of small LLMs, improving their ability to interpret and track user goals throughout a dialogue. These small LLMs are beneficed for their efficiency and deployability in resource-constrained environments.

Traditionally DST research mostly focused on specialized architectures to capture dialogue context Wu et al. (2019b); Heck et al. (2020); Hosseini-Asl et al. (2020); Jacqmin et al. (2022), while recent efforts have shifted toward leveraging LLMs with techniques like synthetic data generation and knowledge distillation to develop small, efficient models Dong et al. (2024b); Finch & Choi (2024); Hu et al. (2022); Wu et al. (2023). Supervised fine-tuning (SFT) has been the dominant approach, aligning predicted dialogue states with ground-truth annotations. While SFT has driven significant progress, recent RL studies show LLMs can improve complex task performance via test-time scaling OpenAI (2024); Muennighoff et al. (2025); Snell et al. (2024); Shao et al. (2024). Similarly, small LLMs works well in domains like mathematics and coding when optimized with RL Zeng et al. (2025); Pan et al. (2025). Despite these advances, the use of RL to tackle DST's unique challenges, particularly for small LLMs, remains largely unexplored.

Applying RL to DST introduces distinct challenges. First, DST datasets, such as MultiWOZ Eric et al. (2020), exhibit an imbalanced *difficulty* distribution ( i.e. with most dialogues being straightforward, a minority being complex due to multi-domain interactions, ambiguous inputs, or intricate slot dependencies). This imbalance affect the ability of RL models to generalize across varying dialogue complexities. Second, the standard DST evaluation metric, Joint Goal Accuracy

(JGA) Budzianowski et al. (2018), enforces a strict requirement of exact slot-value predictions across multiple turns, offering no partial credit. This results in sparse rewards, complicating RL training especially for small LLMs with limited capacity to learn from infrequent feedback. We propose two solutions to enhance RL-based DST for small LLMs:

**Dynamic Difficulty Sampling** A method that uses reward-driven Gaussian sampling to dynamically select training data matching the model's current ability, improving learning efficiency by focusing on moderately challenging examples.

**Weighted Fuzzy Match Reward Function** A function that provides granular feedback by combining slot-level fuzzy matching with difficulty-based weighting. This grants partial rewards for imperfect predictions, with greater emphasis on difficult slots.

In conclusion, our contributions are in three-fold:

1. We address the distinct challenges of applying verifiable RL to DST and propose an effective solution.

2. We introduce an RL pipeline tailored for zero-shot cross-domain DST, enhancing the model's reasoning capabilities in unfamiliar domains.

3. Our experiments demonstrate that the proposed solution achieves state-of-the-art performance in small LLMs settings.

## 2 PRELIMINARY

### 2.1 DIALOGUE STATE TRACKING (DST)

DST maintains a structured representation of user goals in task-oriented dialogue systems, typically as slot-value pairs (e.g., restaurant name, cuisine type). At turn $t$ the dialogue state $b_t = \{(s_1, v_1), (s_2, v_2), \ldots, (s_k, v_k)\}$ where $s_i$ is slot and $v_i$ is value. The dialogue history is $h_t = [u_1, r_1, u_2, r_2, \ldots, u_t, r_t]$ with $u_i$ as the user's utterance and $r_i$ as the system's response at turn $i$. The DST model, parameterized by $\theta$, predicts the state::

$$\hat{b}_t = f_\theta(h_t),$$

For LLMs with test-time scaling capabilities, the output includes both the predicted state and intermediate reasoning:

$$\{think_t, \hat{b}_t\} = o_t = f_\theta(h_t),$$

where $o_t$ is the complete output, and $think_t$ represents the reasoning content.

### 2.2 GROUP RELATIVE POLICY OPTIMIZATION (GRPO)

GRPO Shao et al. (2024) is an advanced reinforcement learning algorithm that enhances the reasoning capabilities of LLMs by evaluating groups of generated responses relative to one another. Its efficiency, demonstrated in models like DeepSeek R1, stems from eliminating the need for a separate value model, unlike Proximal Policy Optimization (PPO) Schulman et al. (2017).

In the context of DST, each dialogue turn is treated as a question-answer pair $(h_t, b_t)$, where $h_t$ is the dialogue history and $b_t$ is the true dialogue state. The policy $\pi_{\theta_{old}}$ generates a group of $G$ candidate responses $\{(\hat{b}_i, think_i)\}_{i=1}^G$, or equivalently $\{o_i\}_{i=1}^G$. A reward function evaluates each predicted state, producing rewards $\{r_i = \text{reward}(\hat{b}_i, b_i)\}_{i=1}^G$ donate as $\{R_i\}_{i=1}^G$. The advantage for each response is computed relative to the group mean:

$$A_{i,t} = r_i - mean(\{R_i\}_{i=1}^G)$$

Recent studies have identified limitations in standard GRPO Liu et al. (2025), including response-level length bias, where longer responses may be favored, and question-level difficulty bias, where performance varies with task complexity. Additionally, in our experiments, for small LLMs (¡3B) the KL divergence term in GRPO's objective can limit exploration and optimization. To address

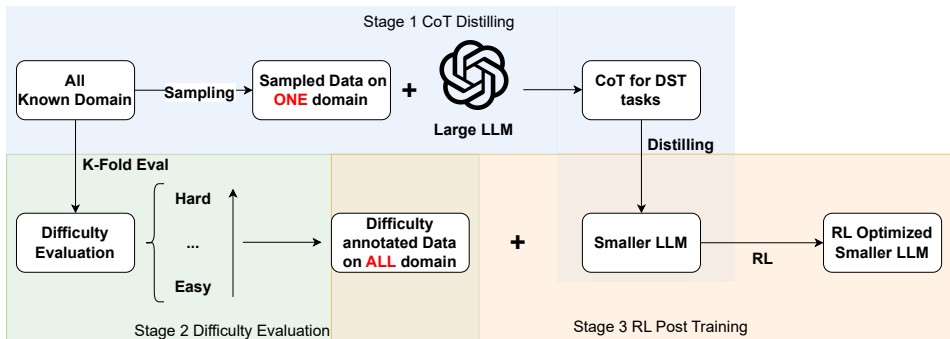

Figure 1: Proposed three-stage RL framework for zero-shot cross-domain DST post-training, featuring CoT distillation, difficulty evaluation, and RL optimization with GRPO.

these issues, we adopt a modified GRPO variant that removes these biases and omits the KL divergence term, enhancing its suitability for small LLMs in DST tasks. The policy is updated by maximizing the following objective:

$$J_{\text{GRPO}}(\theta) = \mathbb{E}_{\substack{(h,b)\sim\mathcal{D}, \\ \{o_i\}_{i=1}^{G}\sim\pi_{\theta_{\text{old}}}(\cdot|h)}} \left[ \frac{1}{G} \sum_{i=1}^{G} \sum_{t=1}^{|o_i|} L_{i,t}(\theta) \right],$$

where $L_{i,t}(\theta)$ is:

$$L_{i,t}(\theta) = \min\left( r_{i,t}(\theta)\hat{A}_{i,t}, \text{clip}(r_{i,t}(\theta), 1-\epsilon, 1+\epsilon)\hat{A}_{i,t} \right),$$

and

$$r_{i,t}(\theta) = \frac{\pi_\theta(o_{i,t}|h, o_{i,<t})}{\pi_{\theta_{\text{old}}}(o_{i,t}|h, o_{i,<t})}.$$

While DST and GRPO provide a foundational framework, applying RL to DST presents unique challenges, including imbalanced difficulty distributions and sparse rewards. In the following sections, we address these challenges through a novel RL pipeline and innovative sampling and reward strategies.

## 3 PROPOSED METHOD

### 3.1 A NOVEL FRAMEWORK FOR CROSS-DOMAIN DST POST-TRAINING

To address difficulty imbalance and improve generalization in zero-shot cross-domain DST, we propose a three-stage reinforcement learning (RL) pipeline to optimize small LLMs (¡3B parameters), as illustrated in Figure 1.

**Stage 1: Chain-of-Thought (CoT) Generation and Distillation.** First, we use a large LLM to generate high-quality Chain-of-Thought (CoT) reasoning for a single domain. This reasoning is then distilled into our small LLM via Supervised Fine-Tuning (SFT) to establish a strong reasoning foundation, which small models struggle to develop from instructions alone.

**Stage 2: Difficulty Evaluation and Data Preparation.** Next, we evaluate the difficulty of the multi-domain dataset to facilitate RL optimization. Using k-fold evaluation with a small SFT model, we ensure a robust and unbiased assessment of dialogue difficulty across the dataset. Difficulty is quantified as the per-turn Average Goal Accuracy (AGA) Rastogi et al. (2020), the ratio of correctly predicted slots to total slots. These difficulty annotations inform the Dynamic Difficulty Sampling strategy (Section 3.2) and Weighted Fuzzy Match Reward Function (Section 3.3), addressing the imbalanced difficulty distribution in datasets like MultiWOZ and enhancing the RL process.

**Stage 3: Reinforcement Learning with GRPO** Finally, we apply the modified GRPO algorithm (Section 2.2) for RL across all known domains. Building on the difficulty annotations from Stage

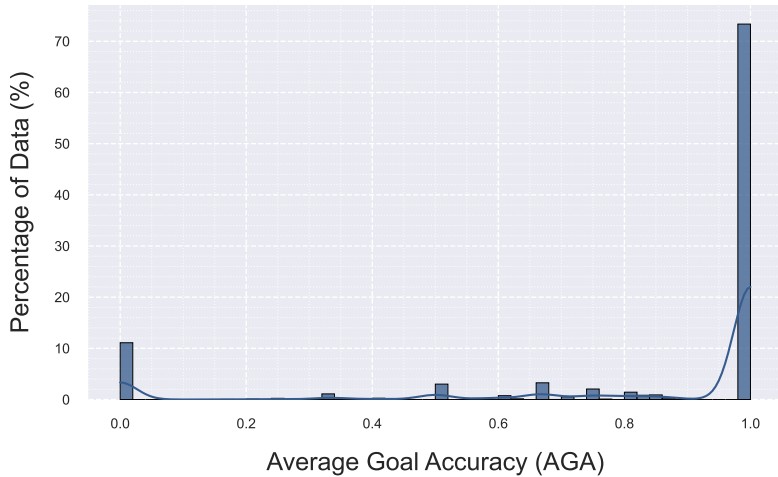

Figure 2: AGA distribution in MultiWOZ, showing difficulty imbalance

2 and overcoming the single-domain limitation of Stage 1. This RL phase enables the small LLM to learn generalized patterns, significantly enhancing its zero-shot performance on unseen domains. The result is an optimized DST system capable of handling new domains effectively.

## 3.2 DYNAMIC DIFFICULTY SAMPLING

In rule-based verifiable RL methods like GRPO, optimization relies on group-level advantages derived from multiple generated responses. However, DST datasets with imbalanced difficulty distributions skew uniform sampling towards easier examples with weaker optimization signals or less benefit from overly difficult samples. We assess the degree pf difficulty using k-fold evaluation, measuring per-turn AGA. Figure 2 illustrates this imbalance in the MultiWOZ dataset.

While static filtering of mid-difficulty samples can initially enhance optimization, as the model's proficiency evolves it introduces bias, since the filter doesn't adapt to the model's changing capabilities. The proposed **Dynamic Difficulty Sampling** strategy adjusts to the model's ability by selecting samples via a Gaussian distribution centered on a target difficulty ($\mu$) with a range ($\sigma$):

$$p_i = \frac{1}{\sqrt{2\pi\sigma^2}} \exp\left(-\frac{(d_i - \mu)^2}{2\sigma^2}\right),$$

where $d_i$ is the sample's difficulty. We adjust $\mu$ and $\sigma$ periodically based on the average reward, shifting the focus to harder or easier samples as performance changes. To ensure multi-domain balance, we apply stratified sampling, maintaining proportional domain representation. This dynamic, balanced approach enhances optimization for DST's imbalance difficulty.

## 3.3 WEIGHTED FUZZY MATCH REWARD FUNCTION

In rule-based RL, reward functions guide models toward optimal performance. For DST, a conventional rewards rely on exact matches between predicted state $\hat{d}$ and ground-truth state $d$, defined as:

$$R(\hat{d}, d) = \begin{cases} 1 & \text{if } \hat{d} = d \\ 0 & \text{otherwise} \end{cases}$$

However, this approach is inadequate for DST tasks. Predicted states in DST comprise multiple slot-value pairs, and partial correctness is common. The exact match reward function assigns a positive reward only when all slot-value pairs are perfectly matched, thereby disregarding any partial progress achieved by the model. Furthermore, this approach encounters difficulties with non-categorical slots, which lack a predefined set of values and allow for open-ended responses. In

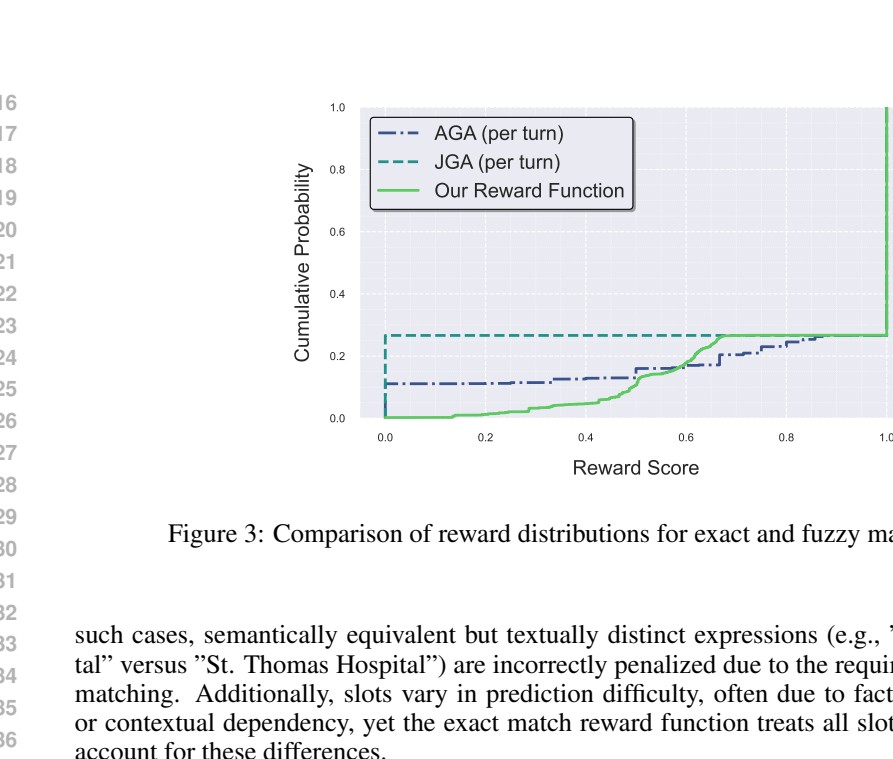

Figure 3: Comparison of reward distributions for exact and fuzzy match functions

such cases, semantically equivalent but textually distinct expressions (e.g., "Saint Thomas Hospital" versus "St. Thomas Hospital") are incorrectly penalized due to the requirement for exact string matching. Additionally, slots vary in prediction difficulty, often due to factors such as ambiguity or contextual dependency, yet the exact match reward function treats all slots uniformly, failing to account for these differences.

Our Weighted Fuzzy Match Reward Function addresses these issues by refining the reward calculation in three ways:

- **Fuzzy Matching for Partial Credit**: We compute a fuzzy match ratio for each slot-value pair. If it exceeds a threshold $\tau$, a partial reward $\delta < 1$ is assigned, enhancing feedback beyond binary matches.

- **Difficulty-Based Slot Weighting**: Leveraging the difficulty evaluation from pipeline Stage 2, we assign weights to each slot based on its error rate. Slots with higher difficulty receive greater weights, directing the model's attention to areas needing refinement and boosting overall robustness.

- **Per-turn Reward Aggregation**: For each slot, we first calculate its individual reward: a full reward of 1 for an exact match, a discounted reward of $\delta$ if the fuzzy match ratio meets or exceeds $\tau$, or 0 otherwise. Next, each slot reward is multiplied by its difficulty weight. These weighted rewards are then averaged across all slots in the turn to produce a single turn-level reward. If not all slots achieve exact matches, we apply a discount factor $\gamma < 1$ to this average, balancing recognition of progress with the goal of achieving complete accuracy.

Formally, the turn-level reward $R_t$ for slots $s_1, s_2, \ldots, s_k$ is:

$$
R_t = \begin{cases} 1 & \text{if } \hat{d} = d \\ \gamma \cdot \left( \frac{1}{k} \sum_{i=1}^{k} w_i \cdot r_i \right) & \text{otherwise} \end{cases}
$$

where $w_i$ is the difficulty weight for slot $s_i$, and $r_i$ is:

$$
r_i = \begin{cases} 1 & \text{if exact match} \\ \delta & \text{if fuzzy match ratio} \geq \tau \\ 0 & \text{otherwise} \end{cases}
$$

As shown in Figure 3, this approach provides nuanced feedback, rewarding partial matches and prioritizing difficult slots, enhancing DST performance in RL frameworks.

# 4 EXPERIMENT

## 4.1 EVALUATION DATASET

We evaluate our approach on MultiWOZ 2.1 Eric et al. (2020) and 2.4 Ye et al. (2022). We report Joint Goal Accuracy (JGA) Budzianowski et al. (2018), the exact match accuracy for all slot-value pairs, and Average Goal Accuracy (AGA) Rastogi et al. (2020), the accuracy of individual slot predictions.

## 4.2 EVALUATION BASELINE

We follow the standard zero-shot cross-domain protocol, training on MultiWOZ with one domain held-out for testing. We benchmark against baselines grouped by parameter count:

- **<1B**: TRADE Wu et al. (2019a), T5DST Lin et al. (2021b), TransferQA Lin et al. (2021a), Prompter Aksu et al. (2023), D3ST Zhao et al. (2022), and CAPID Dong et al. (2024a).
- **1B–100B**: D0T Finch & Choi (2024), SDT Gupta et al. (2022), FNCTOD Li et al. (2024), and LDST Feng et al. (2023).
- **>100B**: IC-DST Hu et al. (2022), RefPyDST King & Flanigan (2023), and ParsingDST Wu et al. (2023). These models are typically closed-source and evaluated in a few-shot setting.

## 4.3 IMPLEMENTATION DETAILS

We use Qwen2.5-0.5B and Qwen2.5-3B Yang et al. (2024) as base models. For Chain-of-Thought (CoT) distillation, QwQ-32B QwenTeam (2025) generates 2,000 reasoning samples each for the hotel and train domains, which are then distilled into the base models via Supervised Fine-Tuning (SFT). We calculate initial difficulty scores using 2-fold validation on a fine-tuned Qwen2.5-0.5B.

For the Reinforcement Learning (RL) stage, we use Dynamic Difficulty Sampling with initial parameters $\mu = 0.75$ and $\sigma = 0.15$, step sizes $\mu_s = 0.05$ and $\sigma_s = 0.01$, and minimums $\mu_{\min} = 0.3$ and $\sigma_{\min} = 0.05$. Reward thresholds are set to 0.7 (0.5B model) and 0.75 (3B model), and fuzzy/partial matching ratios are both 0.8. We use Llama-factory Zheng et al. (2024) for SFT and Verl Sheng et al. (2024) for RL; see Appendix B for full hyperparameters.

## 4.4 EXPERIMENT RESULTS

| Method | Model | Parms | Attraction JGA / AGA | Hotel JGA / AGA | Restaurant JGA / AGA | Taxi JGA / AGA | Train JGA / AGA | Average JGA / AGA |
|---|---|---|---|---|---|---|---|---|
| TRADE | ELMo | 93.6M | 19.9 / 55.5 | 13.7 / 65.3 | 11.5 / 53.4 | 60.6 / 73.9 | 22.4 / 49.3 | 25.6 / 59.5 |
| T5DST | T5 | 60.5M | 33.1 / - | 21.2 / - | 21.7 / - | 64.6 / - | 35.4 / - | 35.2 / - |
| TransferQA | T5 | 770M | 31.2 / 60.6 | 22.7 / 77.8 | 26.3 / 81.7 | 61.9 / 86.5 | 36.7 / 87.2 | 35.8 / 78.8 |
| Prompter | PPTOD | 60.5M | 35.8 / - | 19.2 / - | 26.0 / - | 66.3 / - | 39.0 / - | 37.3 / - |
| D3ST | T5 | 220M | 56.4 / - | 38.2 / - | 38.2 / - | 78.4 / - | 37.7 / - | 46.5 / - |
| CAPID | T5 | 220+60.5M | 40.9 / 69.0 | 31.1 / 72.6 | 31.6 / 69.1 | 65.4 / 83.8 | 34.3 / 65.9 | 40.7 / 72.1 |
| CAPID | T5 | 220+220M | 33.3 / 64.4 | 43.5 / 83.3 | 37.1 / 75.2 | 87.1 / 92.0 | 49.5 / 73.4 | 50.1 / 77.7 |
| Ours | Qwen2.5 | 494M | **65.4** / **81.9** | **54.2** / **90.1** | 52.7 / 85.4 | 84.1 / 94.9 | 48.5 / 83.4 | **61.0** / **87.1** |
| D0T | T5 | 11B | 61.1 / - | 27.6 / - | 64.3 / - | 46.9 / - | 49.7 / - | 49.9 / - |
| D0T | Llama2 | 13B | 66.6 / - | - / - | 67.2 / - | 48.8 / - | 66.5 / - | 58.5 / - |
| FNCTOD | Llama2 | 13B | 62.2 / - | 46.8 / - | 60.3 / - | 67.5 / - | 60.9 / - | 59.5 / - |
| SDT | T5 | 11B | 74.4 / - | 33.9 / - | 72.0 / - | 86.4 / - | 62.9 / - | 65.9 / - |
| LDST | Llama2 | 7B | 75.6 / - | 63.3 / - | | 91.5 / - | 75.0 / - | 75.8 / - |
| CAPID | T5+Llama2 | 7B+220M | **83.6** / **92.6** | **71.6** / **94.2** | **77.5** / 95.3 | 91.2 / **96.0** | **90.0** / **97.8** | **82.8** / **95.2** |
| Ours | Qwen2.5 | 3B | 75.3 / 89.7 | 64.7 / 93.0 | 75.6 / **95.5** | 86.5 / 93.6 | 78.7 / 95.7 | 76.1 / 93.5 |
| IC-DST | Codex | ¿100B | **62.1** / - | 53.2 / - | 54.9 / - | 71.9 / - | 51.4 / - | 58.7 / - |
| FNCTOD | GPT-4 | ¿100B | 58.8 / - | 45.1 / - | 63.2 / - | **76.4** / - | 69.5 / - | 62.6 / - |
| RefPyDST | Codex | ¿100B | **62.1** / - | **56.6** / - | **68.2** / - | 71.9 / - | **76.1** / - | **68.8** / - |

Table 1: Evaluation Results on MultiWOZ 2.1 dataset

To evaluate our zero-shot cross-domain DST approach, we first address the choice of query strategy. Prior methods use either per-domain or per-slot queries. While per-slot queries simplify the task and improve accuracy (e.g., in LDST Feng et al. (2023), DOT Finch & Choi (2024), and CAPID Dong et al. (2024a)), they incur high computational costs. For instance, in a 12-turn hotel domain dialogue

| Method | Model | Parms | Attraction JGA / AGA | Hotel JGA / AGA | Restaurant JGA / AGA | Taxi JGA / AGA | Train JGA / AGA | Average JGA / AGA |
|---|---|---|---|---|---|---|---|---|
| CAPID | T5 | 220+60.5M | 47.9 / 74.3 | 38.7 / 77.1 | 29.4 / 67.9 | 73.3 / 88.2 | 47.9 / 74.4 | 47.4 / 76.4 |
| CAPID | T5 | 220+220M | 22.8 / 59.3 | 31.3 / 79.0 | 39.1 / 78.7 | 89.3 / 93.8 | 56.7 / 76.9 | 47.9 / 77.5 |
| Ours(SFT only) | Qwen2.5 | 494M | 24.9 / 60.1 | 28.4 / 67.5 | 31.5 / 73.7 | 63.2 / 76.7 | 35.4 / 74.1 | 36.7 / 70.4 |
| Ours(SFT with CoT) | Qwen2.5 | 494M | 18.0 / 58.1 | 21.7 / 57.0 | 27.3 / 63.2 | 67.5 / 79.2 | 28.7 / 72.3 | 32.6 / 66.0 |
| Ours | Qwen2.5 | 494M | **69.7 / 84.5** | **54.4 / 90.1** | **54.0 / 89.5** | 85.5 / 95.2 | 49.4 / 83.0 | **62.6 / 88.5** |
| D0T | T5 | 11B | 68.1 / - | 32.0 / - | 72.3 / - | 50.6 / - | 55.8 / - | 55.7 / - |
| D0T | Llama2 | 13B | 76.8 / - | 56.4 / - | 78.8 / - | 54.7 / - | 76.1 / - | 68.6 / - |
| CAPID | T5+Llama2 | 7B+220M | **84.4 / 93.1** | **71.3 / 94.5** | **79.1 / 95.4** | **91.6 / 96.0** | **89.6 / 97.6** | **83.2 / 95.3** |
| Ours | Qwen2.5 | 3B | 79.2 / **94.5** | 65.2 / **95.2** | 75.4 / **95.9** | 85.2 / 94.1 | 79.4 / **95.5** | 76.9 / **95.0** |
| IC-DST | Gpt Codex | ¿100B | 60.0 / - | 46.7 / - | 57.3 / - | 71.3 / - | 49.4 / - | 56.9 / - |
| ParsingDST | Gpt-3.5 | ¿100B | 65.6 / - | 46.8 / - | **67.7 / -** | **80.6 / -** | 62.6 / - | **64.7 / -** |
| RefPyDST | Gpt Codex | ¿100B | **70.9 / -** | **51.2 / -** | 65.6 / - | 67.1 / - | 69.2 / - | **64.7 / -** |

Table 2: Evaluation Results on MultiWOZ 2.4 dataset

with 10 slots, per-slot queries require 60 queries versus 6 for per-domain. We adopt the per-domain strategy for its efficiency, despite its complexity. More details are discussed in Appendix C.

Our results on MultiWOZ 2.1 and 2.4 are shown in Table 1 and Table 2. For models with fewer than 1B parameters, our Qwen2.5-0.5B (494M) model achieves state-of-the-art performance, with an average JGA of 61.0% on MultiWOZ 2.1 (vs. CAPID's 50.1%) and 62.6% on MultiWOZ 2.4 (vs. CAPID's 47.9%). This reflects superior accuracy and adaptability in lightweight settings. For the 1B to 100B range, our Qwen2.5-3B model scores 76.1% JGA on MultiWOZ 2.1 and 77.7% JGA on MultiWOZ 2.4, closely rivaling CAPID's 7B model, matching top baselines with half parameters.

We also compare our RL-based approach with SFT and CoT distillation on the Qwen2.5-0.5B model. Table 2 shows SFT alone achieves 36.7% JGA, and SFT with CoT only reaches 32.6% JGA, both well below our RL method. CoT distillation particularly struggles in unknown domains, highlighting small LLMs difficult to learn CoT from large LLMs. These results demonstrate that RL with test-time scaling outperforms traditional methods, leveraging small LLMs' reasoning potential more effectively for cross-domain DST.

## 5 ABLATION STUDY

### 5.1 HANDING IMBALANCE DIFFICULTY AND SPARSE REWARD

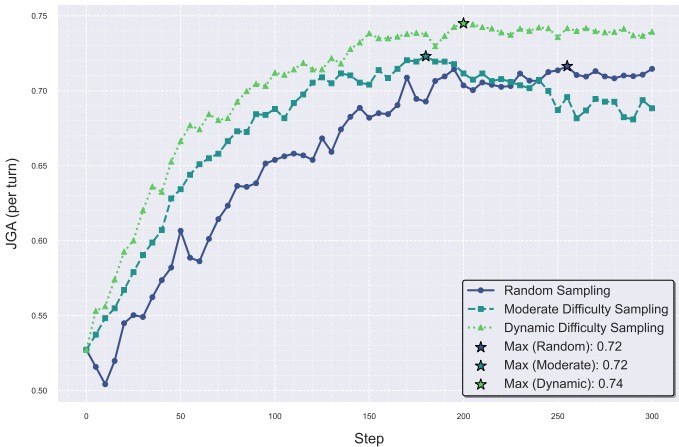

Figure 4: Comparison of Average convergence speed and JGA across different sampling strategies on hotel and train domain

To assess the effectiveness of sampling strategies in RL-based DST post-training, we used the Qwen2.5-0.5B model and the MultiWOZ 2.4 dataset. We conducted zero-shot testing, where the model generalizes to unseen domains without prior training, on the hotel and train domains. For evaluation, we used per-turn JGA, which measures the correctness of all slot-value predictions at each dialogue turn.

| Reward Function | JGA (turn level) | Steps |
|---|---|---|
| JGA (Exact Match) | 0.45 | 500+ |
| AGA (turn-level) | 0.68 | 320 |
| + fuzzy matching | 0.65 | 240 |
| + weighted slot | 0.71 | 180 |
| + partial factor (ours) | 0.74 | 200 |

Table 3: Comparison of different reward function, AGA refers to turn-level partial match

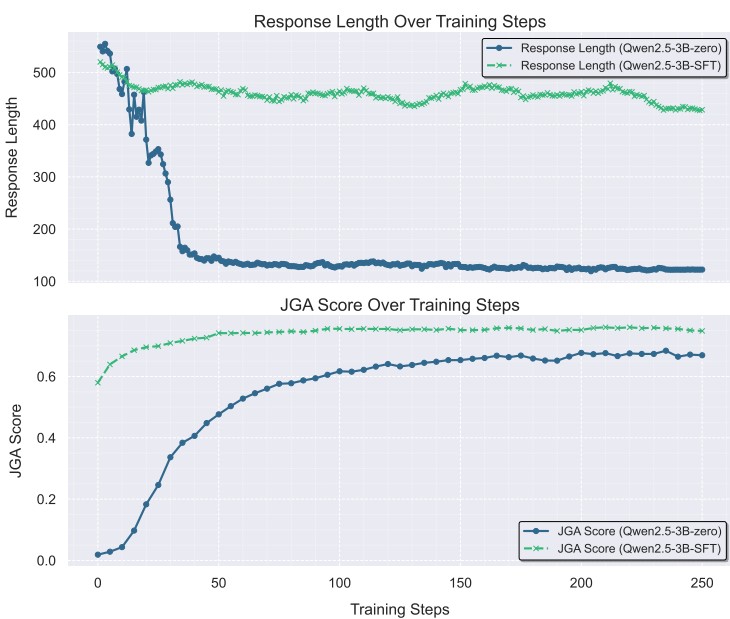

Figure 5: Comparison of reasoning length and JGA between models with and without initial SFT.

We compared three sampling strategies: (1) random sampling (baseline), (2) static moderate difficulty sampling (selecting 8,000 data points with difficulty scores between 0.1 and 0.8), and (3) our proposed dynamic difficulty sampling. Figure 4 shows that our dynamic method converged faster (at 200 steps) and achieved higher JGA than the alternatives. Static sampling outperformed random sampling, confirming the importance of addressing imbalanced difficulty in DST datasets. However, it risked overfitting due to its fixed data pool, while our adaptive approach adjusted to the model's progress, proving more effective.

Next, we explored reward functions' impact on RL-based DST performance, using the same model, dataset, and dynamic sampling. Table 3 summarizes the results after a maximum of 500 training steps. Using JGA directly as the reward led to slow convergence, with gains continuing past 500 steps. Turn-level AGA, which rewards slot-level correctness, markedly improved performance. Adding fuzzy matching—rewarding semantically similar predictions speed up convergence further. Incorporating weighted slots (emphasizing harder slots) and partial credit (for partially correct predictions) resulted in the highest JGA of 0.74 at 200 steps. These enhancements, combined in our Fuzzy Match Reward function, boost both efficiency and accuracy.

Our approach tackles the challenges of imbalanced difficulty and sparse rewards, enhancing zero-shot cross-domain DST with small language models like Qwen2.5-0.5B.

## 5.2 DIRECT RL TRAINING WITHOUT SFT

We investigated whether small LLMs can effectively learn DST from direct RL without an initial SFT stage, a known strategy for larger models Shao et al. (2024). We applied RL directly to the base Qwen2.5-0.5B and Qwen2.5-3B models within a one-shot in-context learning framework.

Our results highlight significant limitations for this approach. The 0.5B model failed to learn, yielding nonsensical outputs. The 3B model reached a JGA of 0.67 but remained well below the 0.79 JGA of its SFT-initialized counterpart. As illustrated in Figure 5, SFT-based models produce longer, more accurate responses (avg. 420 tokens) compared to the limited reasoning of direct RL models (avg. 120 tokens). These findings suggest that SFT is a crucial prerequisite for developing effective DST reasoning in small LLMs.

# 6 RELATED WORK

## 6.1 ZERO-SHOT CROSS-DOMAIN DST

Traditionally, zero-shot cross-domain DST research focused on optimizing model architectures to enable generalization across dialogue domains without domain-specific training Wu et al. (2019a); Lin et al. (2021a); Wang et al. (2021). The emergence of LLMs has shifted the paradigm due to their robust generalization capabilities. Recent studies have prioritized generating informative prompts or synthetic data to enhance zero-shot cross-domain DST performance, typically through SFT in an imitation learning framework. Hu et al. (2022) explored in-context learning for few-shot DST, which can be adapted to zero-shot settings by providing contextual examples. Dong et al. (2024a) introduced context-aware auto-prompting and contrastive decoding to improve LLM performance in cross-domain DST. Finch & Choi (2024) generated diverse synthetic data to make zero-shot DST more adaptable, further leveraging SFT to align models with task requirements. However, the application of RL to optimize test-time scaling and enhance reasoning during inference remain largely unexplored in DST. Our work addresses this gap by investigating RL-driven optimization for small LLMs in zero-shot cross-domain DST, aiming to balance performance and practicality.

## 6.2 VERIFIABLE REINFORCEMENT LEARNING

Reinforcement Learning from Human Feedback (RLHF) Ouyang et al. (2022), particularly using Proximal Policy Optimization (PPO) Schulman et al. (2017), has been shown to significantly improve LLMs' generalization and output quality. Building on this, OpenAI OpenAI (2024) demonstrated that RL can enhance LLMs' reasoning capabilities, implementing test-time scaling to produce more accurate responses. However, these methods require extensive human-annotated reasoning data, which is costly and difficult to obtain, limiting their scalability. To address this challenge, recent research has introduced rule-based verifiable RL approaches such as ReMax Li et al. (2023),RLOO Ahmadian et al. (2024) and GRPO Shao et al. (2024) that reduce reliance on external data. These methods enable base LLMs to self-generate high-quality reasoning tokens, supporting accurate outputs. Despite their success, verifiable RL methods have rarely been applied to DST tasks, particularly with small LLMs. Our study fills this gap by applying GRPO to optimize small LLMs for zero-shot cross-domain DST, demonstrating a scalable and resource-efficient approach.

# 7 CONCLUSION

This study introduces an innovative verifiable RL-based approach to improve zero-shot cross-domain DST using small LLMs. By implementing Dynamic Difficulty Sampling and Difficulty-Weighted Fuzzy Match Reward Function, we tackle issues like imbalanced difficulty and sparse rewards in DST datasets. Optimized with a modified GRPO algorithm, our method achieves top-tier performance among models under 1 billion parameters, rivaling larger models on MultiWOZ 2.1 and 2.4. These findings underscore the viability of small LLMs for efficient, scalable dialogue systems. This work bridges a gap in RL applications for DST and offers a pathway to resource-efficient solutions, enhancing real-world deployment where computational limits are critical.

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

## A  ROLE OF LARGE LANGUAGE MODELS IN THIS STUDY

In this study, LLLMs were integral to both the technical implementation and the final manuscript preparation. For software development, we leveraged an LLM as an intelligent assistant to aid in coding and debugging. Furthermore, to enhance the clarity and quality of this publication, we utilized the LLM to perform comprehensive grammatical checks and proofread the manuscript.

## B  Prompt Template

In this section, we list the prompt we used in our proposed DST RL strategy.

---

**Prompt for DST CoT distilling from large LLMs:**

---

You are an expert in Dialogue State Tracking (DST). Your task is to generate chain of thought for solving provided DST tasks.
## Instructions:
- Generate your chain of thought that helps to arrive at the provided dialogue state in target domain.
- Pay more attention to the difficult and complex dialogue states, and less attention to the easy ones.
- If the dialogue is not related to the target domain(the ground trues dialogue states are all None), please indicate that in your CoT with short explaination.
- The length of CoT should be in 100 to 500 words.
## Input:
- Target Domain:
{domain}
- Dialogue:
{dialogue}
- Related Slots and explanations:
{slots}
- The ground trues dialogue states:
{dialogue_state}
## Output Format:
- Generate answer in <think> tags. Example: <think>Your CoT...</think>
Now, analyze the given content and generate your chain of thought.

---

**Prompt for small LLM generate both CoT and DST results:**

---

For the given dialogue, generate step-by-step reasoning and determine the dialogue state for the domain {domain}.
## Input:
- Target Domain:
{domain}
- Dialogue:
{dialogue}
- Related Slots and explanations:
{slots}
## Output Format:
Output your reasoning in <think> tag and the dialogue state in <answer> tag using following format:
<think>your reasoning..</think>
<answer>slot1:value1,slot2:value2</answer>
Now, analyze the given content and generate your step-by-step reasoning and determine the dialogue state:

---

## C  Detail Hyperparameter

In this appendix, we provide the detailed hyperparameter settings used in our experiments for both Supervised Fine-Tuning (SFT) and Reinforcement Learning (RL) tasks. All experiments were conducted on a 4*H100 cluster.

## C.1 Supervised Fine-Tuning (SFT)

For SFT tasks, we utilized the llama-factory tool as our fine-tuning framework. The hyperparameters are listed in Table 4.

Table 4: Hyperparameters for SFT tasks

| Hyperparameter | Value |
|---|---|
| per_device_train_batch_size | 1 |
| gradient_accumulation_steps | 10 |
| learning_rate | 1.0e-5 |
| num_train_epochs | 3.0 |
| lr_scheduler_type | cosine |
| warmup_ratio | 0.1 |

## C.2 Reinforcement Learning (RL)

For RL tasks, we utilized the verl framework with vLLM as the rollout tool. The hyperparameters are listed in Table 5.

Table 5: Hyperparameters for RL tasks

| Hyperparameter | Value |
|---|---|
| TRAIN_BATCH_SIZE | 512 |
| MAX_PROMPT_LENGTH | 1280 |
| MAX_RESPONSE_LENGTH | 2048 |
| LEARNING_RATE | 1e-6 |
| TEMPERATURE | 1.0 |
| ROLLOUT_N | 5 |

For more details, please refer to our code.

## D Efficiency Analysis of Per-Domain and Per-Slot Query Strategies

In this section, we evaluate the efficiency of per-domain and per-slot query strategies with respect to computational resource utilization, focusing on token usage and the number of queries required. As outlined in Section 4.4, the per-slot query strategy, while simplifying the problem and potentially enhancing accuracy, introduces a substantial computational burden by necessitating individual queries for each slot at every turn. To assess this trade-off quantitatively, we compare a representative per-slot query method, LDST, with our proposed per-domain query model, which is built upon Qwen2.5-0.5B. Two variants of our model are examined: one employing SFT without CoT reasoning, and another incorporating RL post-training with CoT reasoning. The evaluation is conducted using 100 randomly selected dialogues from the MultiWOZ2.4 training set.

The efficiency metrics, including the average number of tokens used for prompts, responses, total tokens, and the average number of queries per dialogue, are summarized in Table 6 below.

Table 6: Efficiency comparison of query strategies in terms of token usage and query count

| Query type | Method | Avg. Prompt | Avg. Response | Avg. total | Avg. #Query |
|---|---|---|---|---|---|
| Per Slot | LDST | 36657.5 | 852.5 | 37510 | 77.5 |
| Per Domain | Ours (DST only) | 3975.75 | 689.75 | 4665.5 | 7.75 |
| Per Domain | Ours (CoT) | 4247 | 3735.5 | 7982.5 | 7.75 |

The results reveal a marked disparity in computational efficiency between the two strategies. The per-slot query approach, exemplified by LDST, consumes an average of 37510 total tokens and requires 77.5 queries per dialogue. In contrast, our per-domain query strategy without CoT reasoning utilizes only 4665.5 total tokens and 7.75 queries, reflecting reductions of approximately 87.6% in token usage and 90% in query count. Even when enhanced with CoT reasoning, which increases token consumption due to the generation of reasoning steps, our per-domain model requires 7982.5 total tokens—still significantly lower than the per-slot strategy.

## E  EXAMPLE RESULT

In this section, we including some fail case from Qwen2.5-0.5B CoT DST model in hotel domain. The incorrect reasoning will be marked as red.

---

LLM hallucination on no existing slots

---

<think>The dialogue revolves around the user requesting a place to stay (hotel) in the north side of town, specifically asking for a hotel with a star of 4, free parking, and free wifi. Let's break down the dialogue to identify the relevant slots for the hotel domain.

1. **Initial Request for Hotel**: The user transitions from requesting a train to requesting a hotel. The key statement is: "I would like a place to stay that is in the north side of town. [SYSTEM]: I can certainly help you with that. There are many accommodations on the north side of town. Do you have any preferences?" This explicitly sets the 'hotel_area' to "north" and the 'hotel_location' to "north". The user specifies "north side of town," which directly sets the 'hotel_region' to "north."

2. **Hotel Domain Establishment**: The user explicitly states: "I would like a place with a star of 4 that includes free parking and free wifi." This directly sets the 'hotel_stars' to "4" and the 'hotel_internet' to "yes." The mention of "free parking" and "free wifi" clearly sets 'hotel_parking' and 'hotel_internet' to "yes."

3. **Hotel Domain Details**: The user specifies "moderate hotel with free wifi and parking." This sets 'hotel_parking' to "yes" and 'hotel_pricerange' to "moderate." The mention of "moderate hotel" indicates a moderate price range, which is captured in 'hotel_pricerange'. The phrase "free wifi and parking" directly sets 'hotel_internet' to "yes" and 'hotel_parking' to "yes."

4. **Hotel Domain Validation**: The user's request for a hotel is explicitly confirmed by the system, and no new slots are added. The user's focus on hotel details (e.g., area, price, parking) is directly mapped to the corresponding slots. The absence of any other slots (e.g., date, number of people, stay) is checked, and all these slots are correctly filled based on the dialogue.

5. **Conclusion**: The dialogue state correctly reflects the user's explicit requirements for the hotel domain, including the slot values specified by the user. No ambiguities arise. The user's request for a hotel is directly addressed, and all relevant slots are properly populated based on the dialogue.</think>

<answer>hotel_area:north, hotel_book-day:NONE, hotel_book-people:NONE, hotel_bookstay:NONE, hotel_internet:yes, hotel_name:NONE, hotel_parking:yes, hotel_pricerange:moderate, hotel_stars:4, hotel_type:guesthouse, hotel_destination:NONE, hotel_contact:NONE </answer>

Misunderstanding on Yes, No and None option.

<think>The dialogue revolves around the user seeking a hotel in the centre of Cambridge for a cheap price range. The target domain is "hotel", so we focus on the hotel-related slots.
1. **Hotel Domain Analysis**: The user's query is strictly about a hotel, so the hotel-related slots (hotel_name, hotel_parking, hotel_book-day, hotel_book-people, hotel_book-stay, hotel_area, hotel_price, hotel_internet, hotel_star, hotel_type) are directly addressed. The user specifies "Alexander Bed and Breakfast" as the hotel, so "hotel_name" is clearly "Alexander Bed and Breakfast".
2. **Hotel Book-Day**: The user selects "Sunday" for the hotel stay, so "hotel_book-day" is "sunday".
3. **Hotel Book-People**: The user confirms "2 people" for the hotel, so "hotel_book-people" is "2".
4. **Hotel Book-Stay**: The user specifies "2 nights" for the hotel, so "hotel_stay" is "2".
5. **Hotel Price Range**: The user specifies "cheap price range" for the hotel, so "hotel_pricerange" is "cheap".
6. **Hotel Area**: The user mentions "centre" for the hotel, so "hotel_area" is "centre".
7. **Hotel Internet**: The user does not mention any internet service provision, so "hotel_internet" is "no".
The user's query is strictly about a hotel, and the dialogue is exclusively about hotel booking. The hotel slot is clearly addressed by the user, and all other slots are directly derived from the user's explicit specification. The absence of any other mentions or requirements related to the hotel (e.g., room type, schedule, or amenities) results in the dialogue state being straightforward.</think>
<answer>hotel_area:centre, hotel_book-day:sunday, hotel_book-people:2, hotel_book-stay:2, hotel_internet:no, hotel_name:alexander bed and breakfast, hotel_parking:NONE, hotel_pricerange:cheap, hotel_stars:NONE, hotel_type:NONE </answer>

---

LLM confused on hotel and guesthouse. the hotel is refer to domain hotel and also for a type of hotel_type slot.

¡think¿The dialogue revolves around the user seeking a cheap hotel. The target domain is "hotel", so we focus on the relevant slots.
1. **Hotel Domain Identification**: The user explicitly mentions "cheap guesthouse" and "cheap hotel" as requirements. The system provides "Alexander Bed and Breakfast" as the hotel, and the user's query is about a cheap hotel. The domain is "hotel", so all slots related to the hotel are addressed.
2. **Slot Analysis**: - **Hotel Name**: The user specifies "Alexander Bed and Breakfast" as the hotel. This directly fills the 'hotel_name' slot. - **Hotel Price Range**: The user specifies "cheap" for the hotel, so 'hotel_pricerange' is "cheap". - **Hotel Book-Day**: The user does not mention a specific day for checking in. The guesthouse is booked for Friday, but the hotel is not specified. The system's response is about the guesthouse, not the hotel. - **Hotel Book-People**: The user booked a guesthouse for 8 people, not a hotel. The hotel request is unrelated. - **Other Slots**: The user's query is about a hotel, not a guesthouse. The parking, restaurant, or other details are not mentioned in the hotel context.
3. **No Other Slots**: The user's query is strictly about a hotel, and the system provides details about the hotel. There is no mention of the hotel's area, booking day, or other parameters. The guesthouse's details (e.g., parking, WiFi) are unrelated to the hotel domain.
The dialogue is strictly about the user seeking a cheap hotel, and the relevant slots are directly addressed. There is no overlap with the guesthouse's details, so all slots related to the hotel are properly populated.¡/think¿¡answer¿hotel_area:NONE,hotel_book-day:NONE,hotel_book-people:NONE,hotel_book-stay:NONE,hotel_internet:NONE,hotel_name:NONE,hotel_parking:NONE, hotel_pricerange:cheap,hotel_stars:NONE,hotel_type:NONE¡/answer¿

