# OpenReview forum: "Zero-Shot Cross-Domain Dialogue State Tracking with Small LLMs: Learning to Think through Reinforcement Learning"
_ICLR.cc/2026/Conference — ICLR 2026 Conference Withdrawn Submission_

### Official Review · Reviewer_DXMd · 2025-10-21

**Soundness:** 3
**Presentation:** 2
**Contribution:** 3
**Rating:** 2
**Confidence:** 2

**Summary:**

This paper explores the use of reinforcement learning, specifically modified GRPO, on top of SFT to fine-tune small LLMs for zero-shot, cross-domain dialogue state tracking (DST). Their contributions include developing two solutions for addressing the challenges of RL for DST--dynamic difficulty sampling (due to the huge difficulty imbalance in the training data) and a weighted fuzzy match reward function to give partial rewards for semantically similar answers and to weight the rewards based on the data sample's difficulty.

**Strengths:**

While my knowledge of recent DST work is admittedly out of date, the paper seems original. The use of RL for DST is interesting and a natural next step, and the authors' implementation of dynamic difficulty sampling and fuzzy rewards based on difficulty are novel and impactful. The ablation studies are great. The overall organization and arguments of the paper are clear.

**Weaknesses:**

There are numerous typos that limit the clarity of the writing. The primary weakness for me is that CAPID, although more than twice as large (i.e., 7B) as the authors' largest model (3B), is clearly superior to the proposed method. I understand that the focus here is on small LLMs, but in order for the experimental results to be truly convincing to the reader, I would need to see an apples-to-apples comparison of the proposed method to CAPID with the same number of parameters.

**Questions:**

Did you invent the GRPO variant, or is there a reference that should be cited at the top of page 3 where the reader can find more info?

Is chain-of-thought reasoning necessary for DST? Your ablation shows SFT only, SFT with CoT, and RL after SFT with COT, but the SFT with CoT was actually worse than SFT only.

A sample of a few of the typos:
- "beneficed" (line 037)
- "small LLMs works well" (line 047)
- "are in three-fold" (line 065)
- "::" (line 083)
- "donate" instead of "denoted" (line 100)
- upside down exclamation point on line 107 before "3B"
- backwards quotes
etc.

---

### Official Review · Reviewer_muPZ · 2025-10-27

**Soundness:** 2
**Presentation:** 3
**Contribution:** 3
**Rating:** 4
**Confidence:** 4

**Summary:**

This paper addresses the challenge of zero-shot cross-domain Dialogue State Tracking (DST) using small language models (<3B parameters) enhanced through reinforcement learning. The authors propose a three-stage framework: (1) Chain-of-Thought (CoT) distillation from large LLMs, (2) difficulty evaluation via k-fold validation, and (3) RL optimization using GRPO. Experimental results showed the proposed approach achieves state-of-the-art performance among models under 1B parameters on MultiWOZ 2.1 (61.0% JGA) and 2.4 (62.6% JGA), with the 3B model rivaling much larger baselines.

**Strengths:**

- The paper introduces Dynamic Difficulty Sampling with Gaussian-based selection and a Weighted Fuzzy Match Reward Function, effectively addressing two key challenges in DST tasks: imbalanced difficulty distribution and sparse reward signals.
- Paper provides thorough ablation studies examining multiple components (sampling strategies, reward functions, and SFT initialization), offering valuable insights into the factors driving model performance.

**Weaknesses:**

- Evaluation is conducted only in MultiWOZ 2.1 and 2.4. Generalization to other task-oriented dialogue datasets (e.g., Schema-Guided Dialogue) is not explored, limiting claims about broader applicability.
- While the approach achieves competitive results within its parameter class, it substantially underperforms 7B parameter models (76.1% vs. 82.8% JGA compared to CAPID 7B+220M on MultiWOZ 2.1). This raises important questions about the practical trade-offs between model efficiency and accuracy for real-world deployment scenarios.
- The model generates an average of 420 tokens per prediction, which poses significant challenges for deployment in latency-sensitive applications such as real-time voice assistants. The paper would benefit from discussing inference optimization strategies or latency-accuracy trade-offs.

**Questions:**

- How does your approach perform on other dialogue state tracking benchmarks beyond MultiWOZ (e.g., Schema-Guided Dialogue)
- How does the proposed 3B model compare to 7B CAPID in terms of inference latency?

---

### Official Review · Reviewer_BLXQ · 2025-10-29

**Soundness:** 3
**Presentation:** 3
**Contribution:** 3
**Rating:** 6
**Confidence:** 3

**Summary:**

The paper targets zero-shot cross-domain Dialogue State Tracking (DST) with small LLMs (<3 B). It proposes a three-stage reinforcement-learning pipeline. Distill Chain-of-Thought (CoT) reasoning from a large teacher into a small model via supervised fine-tuning. Quantify per-slot difficulty across MultiWOZ and use it to build Dynamic Difficulty Sampling (Gaussian centred on current ability, updated by reward). Difficulty-Weighted Fuzzy-Match Reward (partial credit + emphasis on hard slots). Post-train the small LLM with Group Relative Policy Optimisation (GRPO, KL-free). On MultiWOZ 2.1/2.4 zero-shot splits the 494 M model reaches 61-63 % JGA (new SOTA for <1 B) and the 3 B model hits 76-77 %, matching 7-13 B baselines while using 87 % fewer tokens than per-slot methods. Extensive ablations show RL alone fails without SFT; fuzzy + weighted rewards speed convergence; dynamic sampling beats static curricula.

**Strengths:**

1. The paper is well-motivated, the proposed model is concise and easy to understand, directly addressing the problem of concern. Dynamic sampling continuously matches training data to model ability, avoiding easy-sample waste or impossible-hard over-fitting. Fuzzy + difficulty-weighted feedback supplies partial credit for semantically close values and emphasises ambiguous slots, overcoming the classic DST sparse-reward problem.
2. Proposed model achieves new state-of-the-art performance, 0.5B small model can even rival 7 B DST systems.

**Weaknesses:**

1. direct RL from scratch collapses on 0.5 B, so the pipeline still depends on a large teacher.
2. Domain-difficulty estimation is static. Once computed from 2-fold AGA it is frozen, concept drift or new slots are not handled, possibly hurting lifelong adaptation.
3. Fuzzy-match threshold & weight schedule are hand-tuned, risking over-fit to some DST data.

**Questions:**

1. What is the computational budget (GPU hours, CO₂) versus large-model SFT?
2. Can difficulty metrics be learned online or meta-learned for each new client domain?
3. How sensitive are hyperparameters, such as miu, to different annotation data (e.g., MultiWOZ 2.4 vs. SGD)?

---

### Official Review · Reviewer_AnJc · 2025-10-30

**Soundness:** 3
**Presentation:** 3
**Contribution:** 3
**Rating:** 4
**Confidence:** 4

**Summary:**

The paper proposes a Reinforcement Learning-based framework for Dialog State Tracking using small LMs.
The framework features 2 stages: 1) single-domain knowledge+Chain-of-Thought distillation from a larger model to a smaller LLM; 2) multi-domain reinforcement learning of the state tracking policy using Dynamic Difficulty Sampling and Weighted Fuzzy Match reward function.

The technique is evaluated on dialog state tracking datasets MultiWoz 2.1 and 2.4 in comparison to a wide range of previous state tracking approaches, grouped by the model sizes those are based upon (<1B, 1-100B, >100B). The author use Qwen2.5-0.5B/2.5B as target model backbones and QwQ-32B as the teacher for distillation. Evaluation results show that the proposed method outperforms other techniques based on small models and is competitive with larger models.

**Strengths:**

* a new technique for Dialog State Tracking is introduced base on RL with verifiable rewards
* evaluation results on MultiWoz 2.1 and 2.4 show how the technique enables small language models to be a more efficient trackers than a series of prior approaches including models of a larger size

**Weaknesses:**

* the core of the approach has incremental novelty - the DST reward is quite involving. From the RL perspective, it would have been more impressive to see if the technique works with a simpler reward
* tracking dialog state is nowadays assumed implicitly with instruction-tuned LLMs, and DST accuracy can largely be replaced by downstream task accuracy. The paper needs extra motivation for the relevance of the DST task itself
* 2 versions of MultiWoz is still largely one dataset. Given the wide availability of DST datasets from prior works, it makes sense to add more diverse benchmarks for more comprehensive evaluation

**Questions:**

N/A

---

### Note · Authors · 2025-12-09

**Comment:**

We sincerely appreciate the constructive feedback provided by the four reviewers, which highlighted specific shortcomings in our experimental data. Consequently, we have decided to withdraw the manuscript at this time to conduct additional experiments and further substantiate our conclusions.

**Withdrawal Confirmation:**

I have read and agree with the venue's withdrawal policy on behalf of myself and my co-authors.